# Controlling for openness in the male-dominated collaborative networks of the global film industry

**Deb Verhoeven**[1‡]\*, **Katarzyna Musial**[2‡], **Stuart Palmer**[3‡], **Sarah Taylor**[4], **Shaukat Abidi**[5], **Vejune Zemaityte**[6], **Lachlan Simpson**[7]

**1** Faculty of Arts and Social Sciences, University of Alberta, Edmonton, Alberta, Canada, **2** Advanced Analytics Institute, School of Computer Science, Faculty of Engineering and IT, University of Technology Sydney, Sydney, New South Wales, Australia, **3** Melbourne Centre for the Study of Higher Education, The University of Melbourne, Melbourne, Victoria, Australia, **4** School of Global, Urban and Social Studies, College of Design and Social Context, RMIT University, Melbourne, Victoria, Australia, **5** Faculty of Arts and Social Sciences and School of Engineering, University of Technology Sydney, Sydney, New South Wales, Australia, **6** School of Communication and Creative Arts, Deakin University, Melbourne, Victoria, Australia, **7** Research Technology Services, University of NSW, Sydney, Australia

‡ These authors are senior authors on this work.
\* deb.verhoeven@ulberta.ca

**Data Availability Statement:** The network datasets analyzed in this study are available at https://doi.org/10.6084/m9.figshare.11959221.v1.

## Abstract

Studies of gender inequality in film industries have noted the persistence of male domination in creative roles (usually defined as director, producer, writer) and the slow pace of reform. Typical policy remedies are premised on aggregate counts of women as a proportion of overall industry participation. Network science offers an alternative way of identifying and proposing change mechanisms, as it puts emphasis on relationships instead of individuals. Preliminary work on applying network analysis to understand inequality in the film industry has been undertaken. However, in this study we offer a comprehensive approach that enables us to not only understand what inequality in the film industry looks like through the lens of network science but also how we can attempt to address this issue. We offer a data-driven simulation framework that investigates various what-if scenarios when it comes to network evolution. We then assess each of these scenarios with respect to its potential to address gender inequality in the film industry. As suggested by previous studies, inequality is exacerbated when industry networks are most closed. We review evidence from three different national film industries on network relationships in creative teams and identify a high proportion of men who only work with other men. In response to this observation, we test several mechanisms through which industry structures may generate higher levels of openness. Our results reveal that the most critical factor for improving network openness is not simply the statistical improvement of the number of women in a network, nor the removal of men who do not work with women. The most likely behavioural changes to a network will involve the production of connections between women and powerful men.

**Funding:** KM, DV: Australian Research Council Discovery Project DP190101087, "Dynamics and Control of Complex Social Networks", https://www.arc.gov.au/ DV: Shuttleworth Foundation Flash Grant, "Opening up the defensive closed networks of the creative industries", https://www.shuttleworthfoundation.org/ The funders had no role in study design, data collection and analysis, decision to publish, or preparation of the manuscript.

**Competing interests:** The authors have declared that no competing interests exist.

## Introduction

Statistics describing inequitable conditions for women in global film industries have been gathered and circulated for more than 30 years. These statistics have barely deviated despite the development and application of a range of equity policies. In some key creative roles such as film director, the participation rates for women have become marginally worse over time [1].

Statistical analysis of women's participation in various workplaces has typically taken the form of retrospective aggregate description. Instead, this article uses new forms of data analysis in order to assess the effectiveness of different strategies for redressing bias against women. We use data from Australian, Swedish and German film industries to propose, compare and evaluate several approaches designed to increase network openness. Our approach is informed by the conclusions of a major longitudinal study of the film industry which found that "female actors have a higher risk of career failure than do their male colleagues when affiliated in cohesive networks, but women have better survival chances when embedded in open, diverse structures" [2].

Our data describes the formation of teams of filmmakers. It contains not only information about the characteristics of projects and the people involved but, also equally importantly, relational data that enables us to look into the structural connections within and across teams working on film projects. In the absence of formal hiring procedures, collaborative networks are especially important in the film industry. Social network analysis (SNA) provides methods for visualising these group relationships, and, through quantitative measures that characterise network structure, provides methods for identifying strategically important components and participants in the network. It also, therefore, points to ways in which these male-dominated networks can be most effectively "dismantled" or, alternatively, "opened up".

This research rests on two interrelated manoeuvres. Firstly, it flips the object of analysis. If we are going to make these industries a better place for women and other under-represented cohorts, then we need to understand the specific operations of gatekeeping that maintain the dominance of men. The second aspect of the project is to use the data we have collected about specific collaboration networks to propose an innovative course of action to change these male-dominated environments. In this sense, this article deviates from previous descriptive accounts of women's marginalisation in two ways: firstly, by focussing on the social dimensions of industry bias (rather than relying on comparative statistical aggregates), and secondly, by offering an assessment of different strategies for producing change in these networks.

Specifically, this article presents the project's findings on the application of four "what if scenarios" which themselves comprise five different strategies based on common social network phenomenon including "rich get richer" and "small-world" for dismantling domination patterns and behaviours in collaborative networks. All the experiments were run on data representing Australian, German, and Swedish film industries respectively. Data represents film directors, producers, and writers and their collaborations in different productions in the form of a network. Our findings reveal that the most efficient strategy for producing open networks is neither to remove the men who do not work with women, nor to just "add women and stir". Instead, we find that generating relationships between male "key players" and women delivers the most successful outcome for gender equity. Our findings strongly propose a course of action for film industry policymakers that deviates from traditional strategies reliant on retrospective numerical counts of industry membership. Simply adding more women is not the most effective pathway to creating change. Instead, our findings stress the importance of revising the relational networks that underpin the formation of creative teams.

The remainder of this paper is structured as follows: first, we review the literature on gender, gatekeeping, and networks. After that, we describe film industry data used in the analysis and introduce the control mechanisms we used to alter the network. A detailed outline of the experiment set-up and their results follows. Next, we discuss how different ways of controlling the networks influence their openness. Finally, we conclude the paper and offer future research directions.

## Review of literature on gender, gatekeeping, and networks

Gender inequality in the creative or knowledge industries, and particularly the film industries, is persistent and consistent. Industries like film, television, music, and the arts more broadly, are marked by stark, longstanding, and, in many cases, worsening inequalities relating to gender, race, ethnicity, class, age, and disability [3,4,5]. Looking at the film industry alone is instructive. The British Film Institute's Statistical Yearbook records that only 15.7 per cent of films were directed by a woman and 21.1 per cent written by a woman in the UK in 2017 [6]– figures that resonate with Lauzen's annual *Celluloid Ceiling* report auditing the top 250 films made in Hollywood in 2017 [5]. Lauzen's US research is valuable in offering not only a snapshot of the blatant inequalities in key creative roles but, crucially, in highlighting how little these fluctuate year on year. According to her research, in 2017 women comprised only 18 per cent of directors, producers, writers, executive producers, editors, and cinematographers working on the top 250 domestic grossing films. This is virtually the same percentage of women working in these roles 20 years ago (17 per cent in 1998). Only one per cent of 2017's top-grossing films employed 10 or more women in key behind-the-scenes roles, while 70 per cent of films employed 10 or more men. In Australia, the percentage of women working in some key creative roles, such as directing, was lower in 2014 than it was in 1972 [1].

Despite various government policy initiatives that have attempted to improve workplace equity, women, racial and ethnic minorities, and the working class are failing to gain parity of entry to and outcomes within the creative industries, while white, middle-class males fare much better [7,8]. Eikhof and Warhurst have observed that many film industries have shifted from "inhouse" production systems to network processes for producing content [7]. This involves tight, project-based funding and employment practices. In these circumstances, producers typically recruit key creative personnel via personal networks and select creative team members who are already known to them by recommendation or previous experience. Randle et al. have identified the amplification of negative impact barriers that both limit participation and advancement in this production model and contribute to the perpetuation of singular industrial cultures [9].

In this project-based model of film production, project teams are constantly assembled, disassembled, and re-assembled. As film industry work is increasingly organized in peripatetic, team-based projects, equality is generated by those *with* whom an individual works rather than by those *for* whom an individual works [10]. Personal power is understood in terms of relative reputational risk rather than based on managing continuing interpersonal relationships [11]. Especially in project-based labor markets such as film, where recruitment depends on interpersonal networks [12–17], social capital is highly important for getting jobs and structuring the market. While much of the literature on social capital highlights its positive and functional aspects, there is a dysfunctional, "dark" side [18] in the form of social exclusion. If recruitment is largely a result of interpersonal, reputationally derived networks, there is a tendency to exclude and discriminate actors based on ascriptive characteristics, regardless of talent or merit [19–21]. Qualitative research suggests that women in particular suffer from labor markets structured by these informal recruitment practices [22,23].

Some theorists have suggested that this kind of industrial setting produces a "re-traditiona-lisation" of gender roles [24,25]. Furthermore, the informal recruitment processes that pre-dominate in many creative fields can work specifically against women who are unable to call on the protections of gender equity/anti-discrimination regulations found in more formalized and bureaucratically regulated workplaces. Oakley notes that the failure of traditional means of opening up employment to excluded groups, such as "equal opportunity" legislation, at the very least suggests that the problems facing those seeking to integrate the workforce of these looser forms of employment are, if anything, more complex and difficult [26].

Evidence-based research on the persistence of gendered gatekeeping across different peri-ods of industrial transformation is limited [27]. The benefits of longitudinal research con-ducted into "merit-based" industries such as the film industry, in which individual success relies heavily on the accumulation of social capital, include the ability to assess the impact of gendered network effects on career progression [28].

One such study has been conducted by Lutter who undertook a large-scale longitudinal study of the career profiles of film actors between 1929 and 2010 [2]. Lutter identified that women experience a "closure penalty" in cohesive networks but their career prospects improve in networks defined by open structures. Lutter notes that women's risk of failure increases if they work in teams with a higher percentage of males at the managerial level (such as directors and producers). Lutter makes a significant contribution to the literature by demonstrating that it is not team-based project organization *per se* that is detrimental to women's careers but the social composition and structure of the teams themselves that is most important.

## Defining the openness of a network

Looking at the current challenges connected with gender imbalance in various employment environments, the question that arises is what the network of connections or collaborations should look like in order to best address gender equity. To date, there has been little detailed literature on the possibilities for feminist uses of network analysis. Gurumurthy challenges feminists to take up a critical engagement with network theories and argues that the dominant paradigm of network theory in the current historical context needs to be understood by femi-nists for its potential and dangers [29]. Leurs aims to elaborate a feminist ethical principle on data studies, production and use, and a part of her paper is dedicated specifically to the issues related to network analysis [30]. Finally, Smith-Lovin and McPherson discuss whether it is possible to create a network theory based on gender [31]. Another paper exemplifies how net-work analysis may be applied to feminist studies in history and examines organizational affilia-tions of 19th-century women reform leaders in New York State as a case study of relations among social movements [32].

Perhaps most pertinent for this study is research by Lutter, who argues that the critical index of equity outcomes is network openness [2]. Lutter demonstrates that women are less disadvantaged in their careers if their network of connections occurs in an open network structure. Lutter uses three measures to determine the level of team openness and information diversity: (i) *the share of newcomers*–the more newcomers within a team, the better chance that creativity and innovation will thrive and the network will become more diverse; (ii) *indi-vidual exposure to different genre backgrounds* and (iii) *team-based diversity measures*–the last two are diversity measures and Lutter claims that the higher both individual and team-based diversity, the more open a network is. In this study we draw on Lutter's experience, but we also firmly embed our definition of the network openness in the network science that gives us mathematical tools to quantify our research.

Borgatti on the other hand, offers an alternative approach in which a network is implicitly opened (fragmented) through the removal of key players which he identifies using balance of node centrality, betweenness, and group centrality measures [33]. While we use Borgatti's approach as one of the ways to create more open networks in order to compare its effectiveness with alternative strategies, we are very much aware that removing nodes (i.e., individual men) from the film industry network is not a realistic policy scenario. Consequently, we have chosen to define openness as the creation of an enhanced collaborative environment in which everybody has improved chances of participation.

From the perspective of network science, it has been shown that **collaborative network structures are the most effective in terms of knowledge creation when they have a high clustering coefficient and at the same time a short average shortest path** [34]. A high clustering coefficient means that nodes form group(s). A short average shortest path means that from any randomly selected node in a network we can easily (in terms of distance) reach any other node. Together these attributes indicate that the network follows a small-world model [35]. Thus, in our research, we define a set of control mechanisms that, by utilizing network features such as friend-of-a-friend phenomenon, enable us to create a small-world network.

However, at the same time, in order to increase diversity and the "flow of information" through a network, as Lutter suggests [2], **we argue that producers should work with a variety of people and not limit their collaborations to only their familiar and tightly knit groups**. From a network science perspective, this means that we should aim at reducing modularity [36] and assortative mixing [37].

In this work we propose to combined two approaches: (i) one for addressing inequality (Lutter's network openness) and (ii) a second for building strong networks of collaborations (small-world phenomenon). Consequently, we define **an open network as one that follows a small-world model and at the same time aims at reducing modularity and node degree assortativity**. There is a very thin line and natural trade-off between all these conditions. A small-world is characterized, among other factors, by a high clustering coefficient, but, at the same time, we claim that too high modularity (division into isolated groups) is not desirable. This we counter-balance by proposing a short average shortest path resulting from a small-world model.

The simulation framework we outline is open for modifications, so, if any alteration to the definition of openness is needed, it can be implemented by developing new simulation scenarios for new control mechanisms which when applied will result in a new network with certain characteristics.

## Data sources and characteristics

Film industry data for this study was obtained from different sources. Because there is no detailed, standardized, and accessible collection of data for the Australian film industry, we established our own parameters and dataset. German film industry data was supplied by Elizabeth Prommer at the University of Rostock, and Swedish film industry data was provided by the Swedish Film Institute. In order to ensure optimum conditions for comparison, date ranges and attributes were aligned as closely as possible. Table 1 summarises the data. A detailed step-by-step process of data collection and curation can be found at the Kinomatics website [38]. Gender attribution was inferred through a combination of provided information, name-set analysis, and manual checking via Google search for pronoun use (she, he) and gender-specific titles (Mr, Ms, Miss, Mrs).

The three national industries we studied are different sizes and are organised around different business and financing structures. And yet they bear remarkable statistical similarities in

**Table 1. Film industry data.**

|  | Total no. of producers (unique people) | Average size of creative team | % of male producers who worked with 0 women | Percentage of male producers who worked with 0–1 woman | Percentage of projects with women in minority (unique people) | Percentage of projects with women in minority (roles) |
|---|---|---|---|---|---|---|
| Sweden (2006–16) | 304 | 3.5 | 46 | 73 | 71.6 | 76.5 |
| Germany (2006–16) | 1,446 | 3.7 | 45 | 75 | 77.8 | 79.0 |
| Australia (2006–15) | 344 | 3.7 | 42 | 75 | 65.9 | 78.0 |

terms of gender dynamics at the aggregate level. Especially notable are the percentage of producers who employed–across the entire sample period–none, or just one woman on any of their creative teams (Table 1). Simply examining the data at this level gives the impression that gender disparity is not only intrinsic to individual film industries, it is globally accordant. However, detailed examination of these gendered dynamics as a social network strongly suggests otherwise.

### Review of social network analysis characteristics

Social network analysis is a very powerful approach for evidence-based policy as it enables us to understand not only the individuals and their behavior but also the relationships between people and their dynamics. The results enable a deep understanding of various social phenomena such as the key and influential members of a network and the conditions that foster their emergence.

In this study, we try to understand how different mechanisms may influence network structures. We will show that through the application of control mechanisms, it is possible to describe a network in which collaboration thrives and the network is more open. In order to understand how different control strategies influence network function, we need to monitor network characteristics and their dynamics over time (simulation runs). Below we present the main characteristics that we calculated to better understand the effects that different control methods cause.

**Degree.** Node degree indicates the number of relationships that a given node in a network has. Nodes with a very large number of connections are called hubs in the network whereas those with a very small number of connections are usually at the periphery of a system. Node degree distribution indicates how many nodes with a specific degree there are in a network.

**Density.** Network density is one of the most intuitive network measures and it shows the extent to which a network is connected. Density ($D$) is the ratio between the number of existing links and all possible links given the number of nodes:

$$D = \frac{2e}{n(n-1)}$$

where $e$ is the number of edges/connections in a network and $n$ is a number of nodes in a network. The range of density is between 0 and 1, where 0 means that a network has no connections and 1 means that a network is fully connected.

**Clustering coefficient.** Clustering coefficient ($CC$) indicates the degree to which nodes in a graph create densely connected clusters [35]. The $CC$ for a network is calculated as an average clustering over all nodes in a network. For each of the nodes (vertex) $v$, its local clustering

coefficient can be calculated by:

$$C_v = \frac{number\ of\ triples\ connected\ to\ v}{number\ of\ all\ possible\ triples\ connected\ to\ v}$$

and then the *CC* for a network can be calculated as:

$$CC = \frac{1}{n} \sum_{v=1}^{n} C_v$$

where *v* is the number of nodes in a network. The range of *CC* is [0;1]. High *CC* indicates that a network is highly clustered, and that people tend to create cliques. Connected networks with high *CC* are also dense networks.

**Average shortest path.**   Shortest path indicates how far two given nodes are from each other. If any two nodes in a network are close to each other, it means that it is easy to reach any node in a network from any other given node. Such structures are very robust and well connected. The average number of the shortest paths (*ASP*) between each pair of vertices is calculated as:

$$ASP = \frac{1}{n(n-1)} \sum_{i \neq j} d(v_i, v_j)$$

where *n* is the number of nodes and $d(v_i, v_j)$ is the shortest path between $v_i$ and $v_j$. The smaller the *ASP*, the easier it is to reach any node in a network.

**Assortativity.**   Assortativity is an important measure as it shows to what extent nodes with similar characteristics are connected to each other. Degree assortativity coefficient (*r*), first defined by Newman [39], is calculated as:

$$r = \frac{\sum_{jk} jk(e_{jk} - q_j q_k)}{\sigma_q^2}$$

where $e_{jk}$ is the joint-remaining degree probability for remaining degree *j* and remaining degree *k* (remaining degree of a node is equal to the degree of that node minus one); $q_k = \frac{(k+1)p_{k+1}}{\sum_j jp_j}$ is the normalized distribution of the remaining degree $D_r$ of a randomly selected node; and $\sigma_q$ is the standard deviation of $q_k$. The range of *r* is [-1;1]. A positive value of *r* indicates a correlation between nodes of similar degree, while a negative value indicates a relationship between nodes of different degree. When *r* is zero, it means a network is non-assortative.

**Modularity.**   Modularity measures the strength of the division of a network into modules [36]. Modularity is defined as a fraction of the edges that are part of the given modules/communities minus the expected fraction if edges were randomly distributed and this can be formulated as:

$$Q = \frac{1}{2m} \cdot \sum_i \sum_j (A_{ij} P_{ij}) \cdot \delta(g_i, g_j)$$

Where $A_{ij}$ denotes the links between node *i* and *j* (1 means that a link from *i* to *j* exists and 0 otherwise); $P_{ij}$ denotes the expected number of links between *i* and *j*; $g_i, g_j$ show which community nodes i and j belong; $\delta(x, y) = 1$ if $x = y$ and *m* is the total number of links in the network.

The range of modularity is [-1;1]. High positive modularity means that the nodes in the same modules/communities are very well connected, and nodes in different modules/communities are sparsely connected. Modularity is used in the process of community detection, and it enables us to create a hierarchy of communities from the whole network to the individual nodes.

**Borgatti fragmentation.** Drawing from the literature on the use of social network analysis to characterize criminal networks and identify key nodes ("players") whose removal would disrupt the network (i.e., [33,40–42]), we investigated the network of male-only producers and other creatives in the film industry. We investigated the impact of key players in these networks, and the hypothetical impact of removing different key players. We used Borgatti's network fragmentation factor ($F$) [33] as a quantitative measure of network disruption:

$$F = 1 - \frac{\sum_k s_k(s_k - 1)}{n(n - 1)}$$

where $s_k$ is the size of component $k$ and $n$ is the number of nodes. In this equation, the $F$ value is 0 when there is no fragmentation in a network (all nodes are connected in a single component) and is 1 when all nodes in a network are isolated.

## Data and extracted networks description

Based on the gathered data, a collaboration network of connections between different creative team members (directors, producers, and writers) was extracted for each country (Figs 1–3). Relationships were created between a source node (always a film producer) and target node (any or all of a director, producer, or writer) who worked with the source node. This definition of a "creative team" conforms to government agency definitions [1]. Basic characteristics of all the networks were calculated and are presented in Table 2.

Most of the network characteristics are very typical for large-scale social networks. All networks are very sparse (density is very low) and they all exhibit power-law node degree distribution (see Figs 4–6). Additionally, all networks are disconnected, meaning that the average shortest path cannot be calculated. They all have many disconnected components, with Germany leading in this statistic with 316 components. However, on average, Australia has the smallest number of nodes in a component, with only seven. The percentage of women in all of the networks is much smaller than men (Sweden and Australia– 31% and Germany– 25%).

The Swedish and German networks have a much lower clustering coefficient than the Australian network, meaning that the former networks have fewer triads (connections between three nodes) than the Australian network. This, in turn, means that the Australian network is closer to a small-world network than the others. In the Swedish and German networks, we see more one-to-one collaboration than in the Australian network, while in the Australian network the friend-of-a-friend phenomenon is present.

Looking at assortativity (based on degree), the Australian network nodes have the tendency to connect with other nodes that have a similar node degree, while the Swedish and German networks do not exhibit this behaviour. "Big boys" tend to stick together only in the Australian network. Borgatti fragmentation and modularity are high for all networks, meaning that, in cases of community structures, the networks tend to create disjointed communities.

To summarize, all networks consist of components that are not highly interconnected to each other, so there is minimal collaboration between constituent communities. The networks feature power-law node degree distribution in which the nodes with the highest number of connections are male.

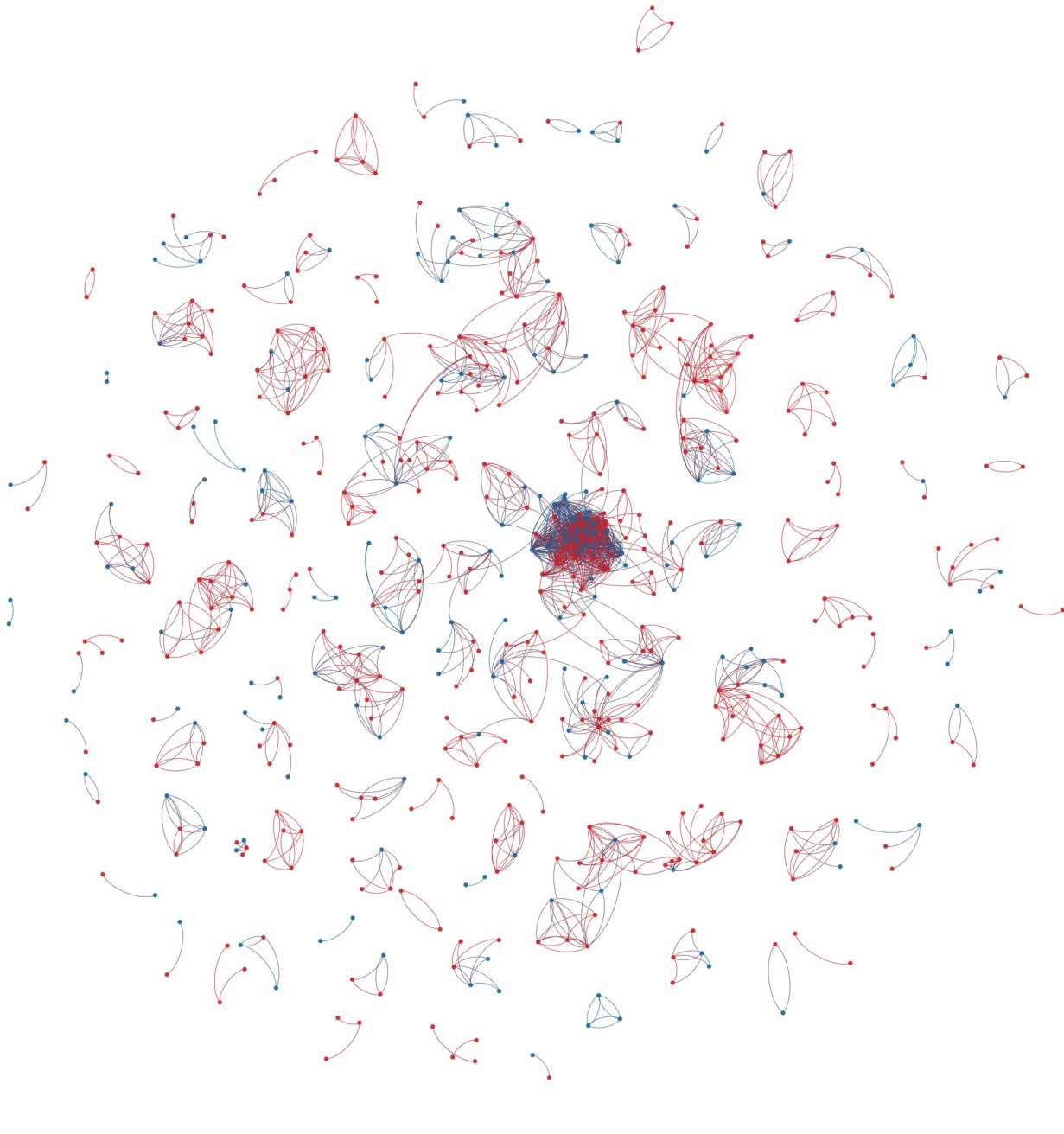

**Fig 1. Australian feature drama production (2006–2015) colour coded by gender.** Red nodes are men, Blue nodes are women. Edge direction from the source (Producer) node to the target (other creatives) node.

## Control mechanisms for increasing network openness

There is a great interest in controlling complex networks, as this has a great potential to make a direct impact on tackling important social challenges. Fundamental research in the area of controlling complex networks has a wide range of applications, one of them being changing human behaviour. Gender imbalance, minority marginalisation, and organised criminal behavior represent only a few challenges with which network science can assist.

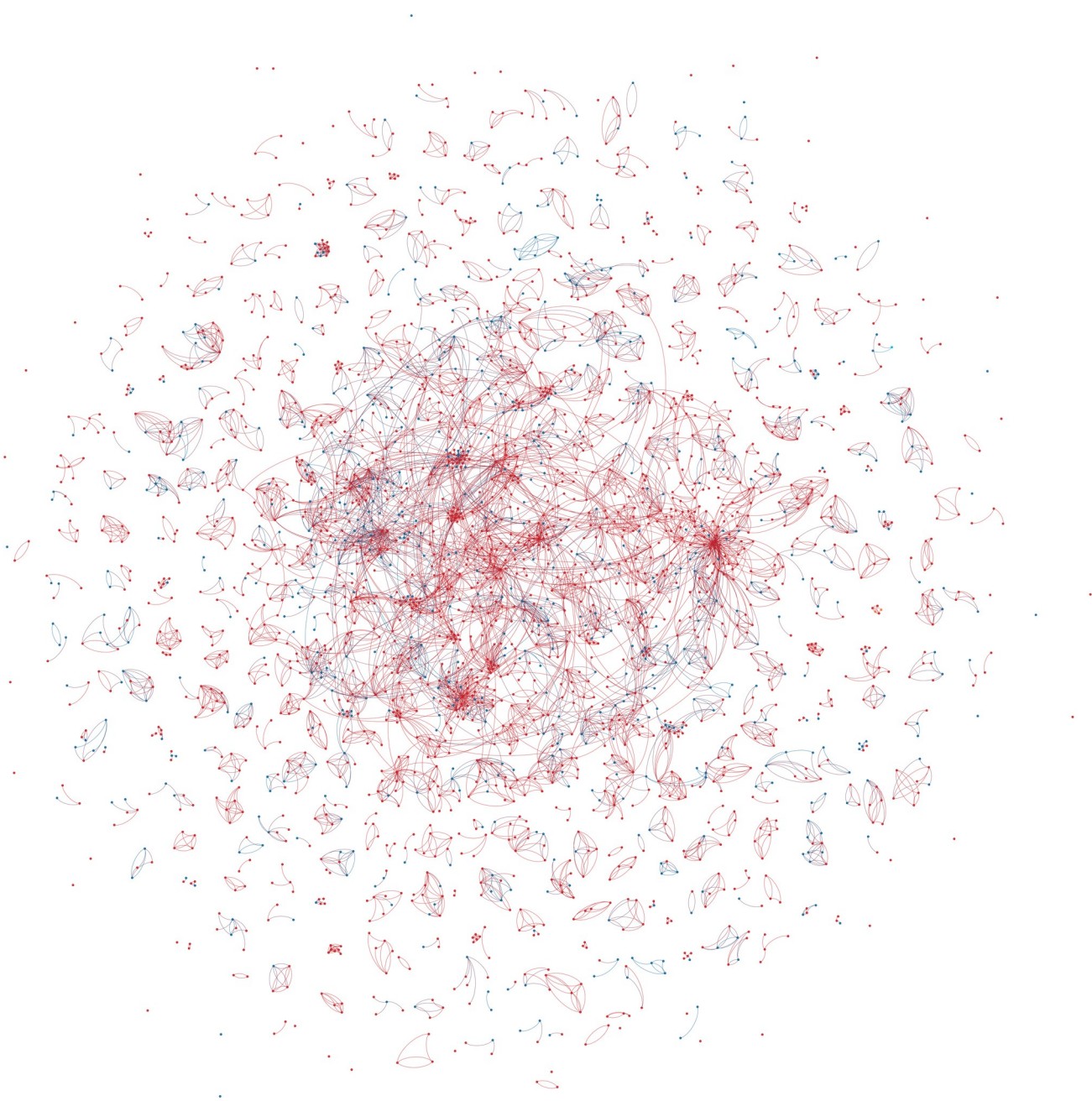

**Fig 2. German feature drama production (2006–16) colour coded by gender.** Red nodes are men, Blue nodes are women. Edge direction from the source (Producer) node to the target (other creatives) node.

In this study, we focus on collaborative networks where gender imbalance is the critical issue. As mentioned before, to increase the diversity and build equity, and at the same time achieve highly collaborative environments, we believe that we should create an open network. In order to build such an open network, we propose and develop a number of "what-if scenarios" that we apply to the networks which represent film industry collaborations between producers, writers, and directors in three different countries: Australia, Germany, and Sweden.

Using control mechanisms to alter the structure of networks in different ways, we investigate which most effectively produces open networks and breaks systemic male domination. The control mechanisms that we use are introduced below.

## Control Mechanisms (CM): Building up a network

**CM1: Friend of a Friend (FoF) approach.** This approach is based on the idea of adding nodes to a network using specific control mechanisms that over time may lead to permanent

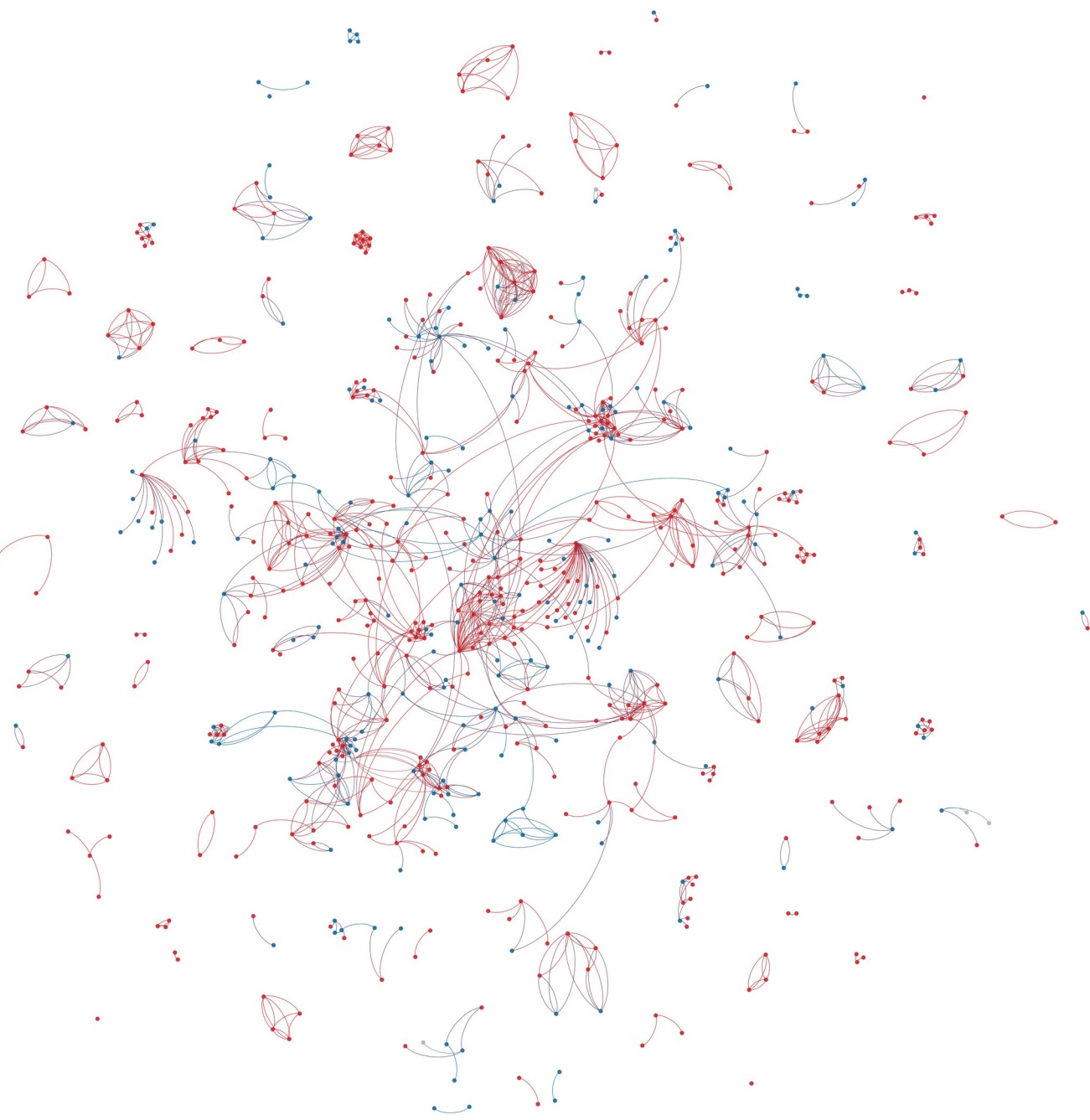

**Fig 3. Swedish feature drama production (2006–15) colour coded by gender.** Red nodes are men, Blue nodes are women. Edge direction from the source (Producer) node to the target (other creatives) node.

**Table 2. Characteristics of extracted networks.**

| Country | No. nodes | No. edges | No. of women | % of women | No. of men | Density | Av. clustering coefficient | Av. degree | Modularity | Assortativity | Borgatti degree fragmentation | No. components | No. of nodes per component |
|---|---|---|---|---|---|---|---|---|---|---|---|---|---|
| Sweden | 687 | 1,044 | 212 | 31 | 475 | 0.002 | 0.083 | 1.520 | 0.908 | -0.154 | 0.924 | 64 | 11 |
| Germany | 2,756 | 5,171 | 679 | 25 | 2,077 | 0.001 | 0.055 | 1.876 | 0.925 | 0.000 | 0.934 | 316 | 9 |
| Australia | 611 | 1,463 | 187 | 31 | 424 | 0.004 | 0.221 | 2.394 | 0.805 | 0.672 | 0.973 | 91 | 7 |

behavioural changes in the whole network. One such mechanism is to encourage men who work only with men to change their behaviour and begin working with women. In response to this, we developed a method based on a well-known social phenomenon in which "a friend of my friend is also my friend". In this way, we can propose connections between men who have never worked with women and women who are structurally close to those men, such that the distance between them in a social network is short. As this strategy is based on a well-known social phenomenon, we believe that it will be relatively easy to introduce in a real-world scenario.

**CM2: "Rich get richer"–skewed preferential attachment.** Another approach to build up a network is to use a well-known phenomenon of "rich get richer" [43]. It means that nodes which are well-connected will create even more links in the future. We propose to alter this mechanism in a way that will encourage men who have worked only with men to change their behaviour and begin also working with women. Thus, in the film producer networks we study, this mechanism creates relationships between male producers and randomly selected women. A producer will be selected using preferential attachment mechanisms such that the probability that he will be selected equals the number of his connections divided by the sum of all connections in the network. Preferential attachment is a phenomenon that occurs in real-world

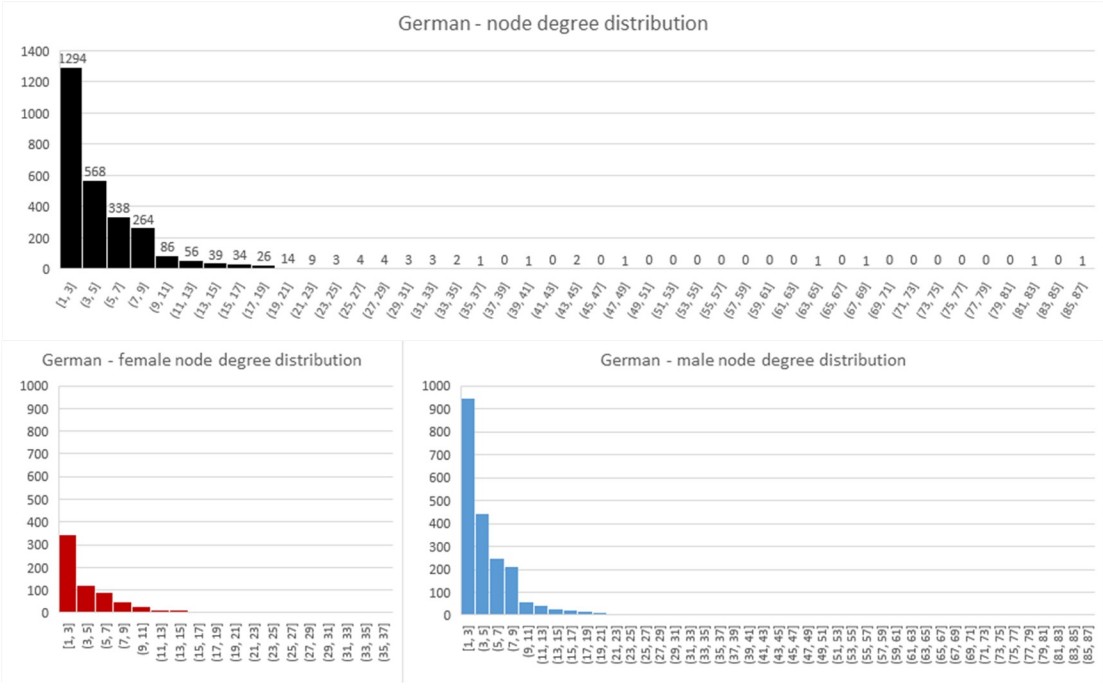

**Fig 4. Node degree distribution for German network.**

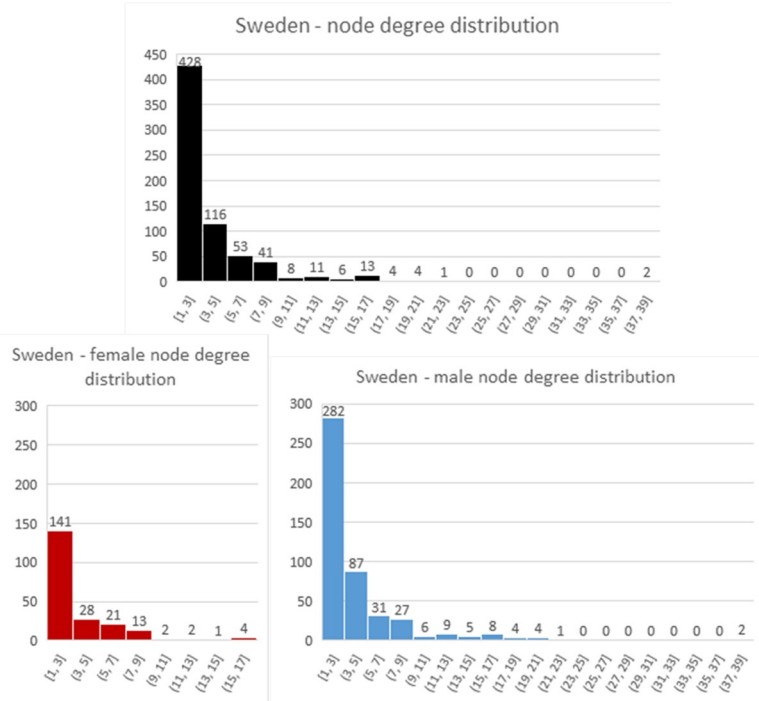

**Fig 5. Node degree distribution for Swedish network.**

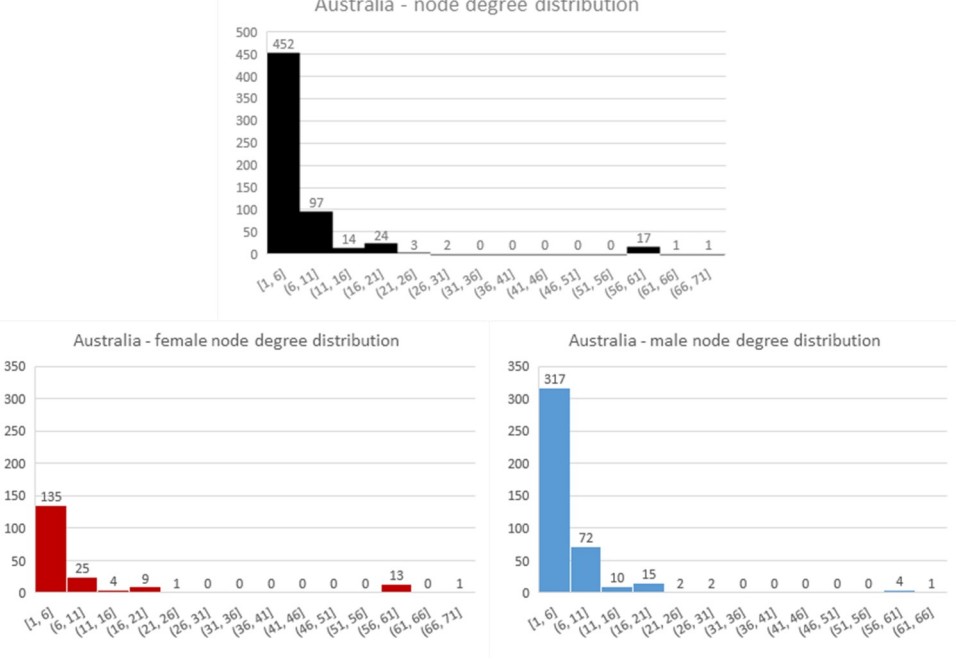

**Fig 6. Node degree distribution for Australian network.** In this chart, nodes with node degree higher than 50 took part in one "compendium" project that involved multiple producers and teams.

networks and we expect that highly connected producers will have increasingly more connections over time, so there is potential to introduce this strategy in real-world cases.

**CM3: Random–skewed random growth.**   In this strategy relationships are created between a randomly selected male producer and randomly selected women. This idea stems from the random graph theory [44], but we change it such that nodes, for which the relationship is created, have some predefined attributes; one is a woman and the other is a male producer. This is yet another method to create more relationships between male producers and women but is not underpinned by any social phenomenon, meaning that it will not be easy to realise in the context of the real world.

## Control Mechanisms (CM): Reducing a network

**CM4: Removing highly connected male producers.**   In this final approach, we take the original network and remove male producers with the highest degree (and all their connections). We repeat this process until no male producers exist in the network. This is an extreme case of control mechanisms and one that may not be possible to implement to its full extent in the context of the real world.

## Experiment set-up

All experiments were performed in a simulated environment built to test the proposed control mechanisms. For each strategy, we present an experimental set-up including all parameters, settings, and the step-by-step simulation process.

### Strategy 1: Friend of a Friend approach–version 1 (FoF v1)

For the Friend of a Friend approach, we propose two different control mechanisms. Version 1, described in this section, adds a specific number of links between selected male producers and **no_links_to_be_created** women who are the closest to them in terms of path length.

*Simulation process*:

a.  Take the original network

b.  Find all men (producers) working with **no_of_women_worked_with** women or less than that and create a table with their IDs and node degree (number of their connections)– **men_set (id, degree)**

c.  For each man in **men_set**, calculate the shortest path to all women in the network

d.  For **no_men** men with the highest degree from **men_set**, create **no_links_to_be_created** links with women closest to them

e.  Repeat steps B-D until no new relationships can be created

Experiment setting:

a.  no_of_women_worked_with = {0, 1, 2}

b.  no_men = {0 (All)}

c.  no_links_to_be_created = {1,2,3,4,5}

Please note that all combinations of parameters were tested and a total of 15 simulations were run for this strategy.

## Strategy 2: Friend of a Friend approach–version 2 (FoF v2)

Version 2 of Friend of a Friend approach adds new links between selected male producers and all women, where the maximum shortest path between them is no longer than *n*.
*Simulation process:*

1. Take the original network

2. Find all men (producers) working with **no_of_women_worked_with** women or less than that and create a table with their IDs and node degree (number of their connections)–**men_set (id, degree)**

3. For each man in **men_set**, calculate the shortest path to all women in the network

4. For **no_men** men with the highest degree from **men_set**, create all the links between those men and women where the shortest path equals to or is less than **shortest_path_value**

5. Repeat steps B-D until no new relationships can be created

   Experiment setting:

a. no_of_women_worked_with = {0, 1, 2}

b. no_men = {0 (All)}

c. shortest_path_value = {2,3,4,5}

   Please note that all combinations of parameters were tested and a total of 15 simulations were run for this strategy.

## Strategy 3: Rich get richer–skewed preferential attachment (RGR)

For each experiment setting from (i) Friend of a Friend–version 1 and (ii) Friend of a Friend–version 2, we ran an experiment where the same number of links was created on the input network but using preferential attachment model, so the results are comparable. If in (i) or (ii) *n* edges were created in a given experimental setting, then *n* connections using preferential attachment model were created in the original network and this network was compared with those coming from (i) and (ii).
*Simulation process:*

a. For each iteration performed in (i) or (ii), take the number of edges created in a given iteration

b. From the current network, randomly select a node–this will be the target node

c. For each producer, calculate their node degree

d. For each producer, calculate the probability that a given node will be selected. Probability is (node_degree_of_a_given_node / sum(node_degrees_of_all_nodes)

e. From the list of producers, select a source node according to the probability calculated in step D

f. Repeat steps B-E until the number of needed edges for a given iteration is reached

   Please note that all combinations of parameters were tested and a total of 15 simulations were run for this strategy.

### Strategy 4: Random–skewed random growth (RAN)

For each experiment setting from (i) Friend of a Friend–version 1 and (ii) Friend of a Friend–version 2, we ran an experiment where the same number of links were created on the input network but using a random model, so the results are comparable. If in (i) or (ii) *n* edges were created in a given experimental setting, then *n* random connections were created on the original network and this network was compared with those coming from (i) and (ii).

*Simulation process:*

a. For each iteration performed in (i) or (ii), take the number of edges created in a given iteration

b. From the current network, randomly select a node–this will be the target node

c. From the current network, randomly select a node that is a producer–this will be the source node

d. Create relationships between the selected source and target nodes

e. Repeat steps B-D until the number of needed edges for a given iteration is reached

Please note that all combinations of parameters were tested and a total of 15 simulations were run for this strategy.

### Strategy 5: Removing the highly connected male producers (REM)

For each *n* = {0, 1, 2, 3, . . ., *k*}, where *k* = the number of male producers, repeat the following steps:

a. *N = 0*

b. Take the original network

c. Remove the *n+1* male producer with the highest degree (and all his connections)

d. Return the resulting network; *n = n+1*

e. Repeat steps B-D on the original network until all producers are removed

There will be *k* steps in the simulation.

## Results, analysis and discussion

### Influence of control mechanisms on network characteristics

This section shows the results of the simulations and discusses the impact of proposed control mechanisms on network structure expressed in networks characteristics. To reiterate, the following naming convention for our results is used:

- FoF–Friend of a Friend procedure (v1 and v2 for version 1 and version 2 of the approach, respectively)

- RGR–rich get richer procedure of creating links

- RAN–random procedure of creating links

- REM–removal of male "key players"

In the plots: CM-vx-no_of_women_worked_with-no_men refers to a specific experimental setting for a specific Control Mechanism (CM), for example, FoF-v1-1-0 means the simulation

was run using version 1 of Friend of a Friend approach where connections were added for all male producers who have worked with one or no women.

## Results: Modularity and clustering coefficient

In network science, modularity and clustering coefficients are the metrics that enable us to assess the extent to which the network structure exhibits community structure and, in turn, supports collaborative behaviour. Adding new links to a network brings the modularity down, meaning that the cliques that existed in the networks are no longer isolated (Fig 7). The more links we added in our experiments, the bigger drop in modularity value we observed.

Additionally, the fact that the clustering coefficient goes down (Fig 8) means that newly created connections cross the boundaries of the communities and the teams become more open. If we keep adding nodes, the clustering coefficient will again increase as the cliques will be already connected (the intended outcome), and then the network will build up, so its density and average clustering coefficient grow.

Note that the drop in modularity and average clustering values are stronger for random (RAN) and rich get richer (RGR) mechanisms, as these two mechanisms can bridge the gaps between disconnected network components, which none of the FoF versions can achieve. This shows that there is a potential for the combination of two or more control mechanisms to achieve a more open network. The most realistic model for producing a more open network is FoF, which works at the local level of single connected components. The next most realistic model is RGR, which works on the global network. Thus, potentially bringing the local- and global-level mechanisms together might result in even further decay of modularity and average clustering coefficient.

## Results: Degree assortativity

Looking at the Australian network (Fig 9), the assortativity drops as we add more and more links, meaning that nodes start mixing together and that nodes are more often connected with

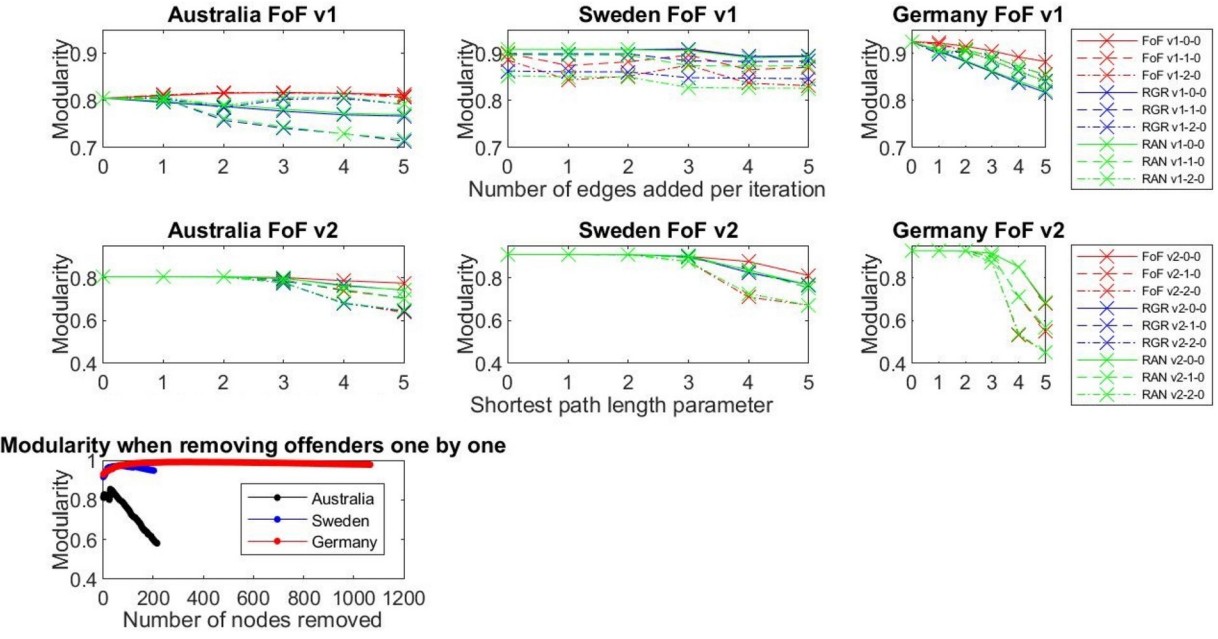

**Fig 7. Modularity for different experiment settings.** For v1 plots, the x-axis is **no_links_to_be_created** and for v2 plots the x-axis is **shortest_path_value**.

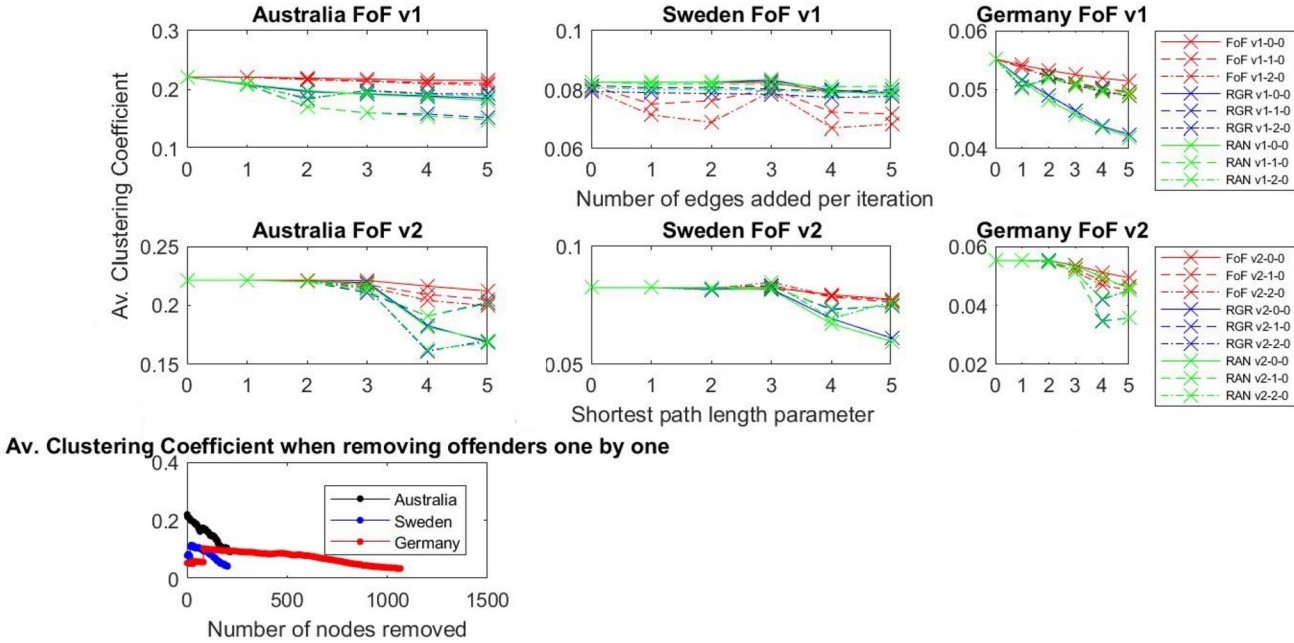

**Fig 8. Average clustering coefficient for different experiment settings.**

other nodes containing different degrees. In the case of Australia, the most effective strategies (RGR, RAN, FoF) are based on the idea that links are added between male producers who worked with zero, one, or two women, so that the largest number of connections is added. This shows that it is not only important to target those male producers who do not work with women at all but also to facilitate the collaborative behaviours of other male producers and encourage them to work with women more often. These results show that using this strategy increases the diversity of connections in the network. This may be an indication that the proposed control mechanisms opened the network and broke the well-connected cliques as well as built a more heterogeneous network.

For the German and Swedish film industries (Fig 9), the assortativity does not drop in the sae systematic way as it does in Australia. However, the assortativity was not high in those networks in the first place. Moreover, as the German and Swedish networks are sparser than the Australian network, assortativity for those two networks does not offer meaningful results.

## Results: Borgatti coefficient

FoF strategies do not give us what we want because the network is very fragmented and adding edges within the fragments does not change the Borgatti measure (Fig 10). With RGR and RAN approaches we can see that if we bridge those fragmented communities and add connections between them, Borgatti declines. Removing nodes (identified as "men who do not work with women") with the highest node degree one by one (REM) does not influence the Borgatti measure a great deal and is not an effective strategy for improving openness in the network.

## Analysis and discussion

**Modularity and Clustering coefficient decrease** as we add more links (FoF approach performs in a similar fashion as RGR and RAN), indicating an opening of the network in the sense that cliques are not so closed. As we mentioned before there is a trade-off between

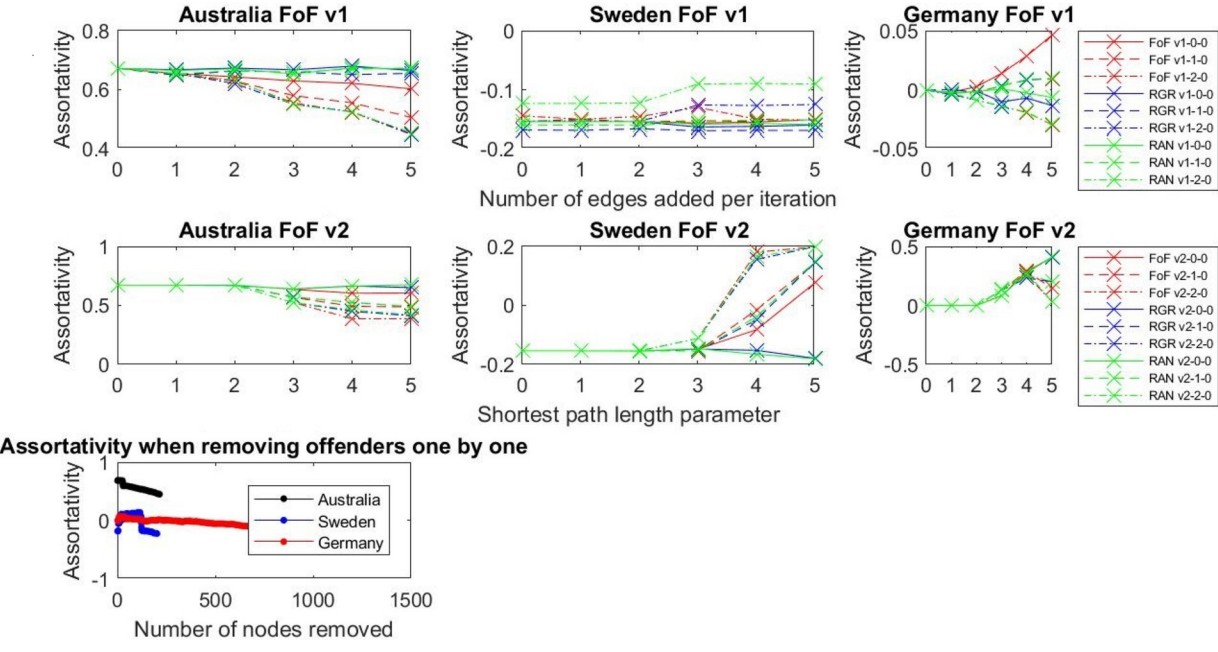

**Fig 9. Degree assortativity for different experiment settings.**

decreasing modularity and enhancing the small-world character of the network, which in traditional sense means increasing the clustering coefficient and decreasing the average shortest path. This trade-off has to be carefully managed as we want to decrease modularity to split men-only cliques, identified in the data, and, at the same time, keep the small-world character of the network. Decreasing modularity also decreases the clustering coefficient. However, a

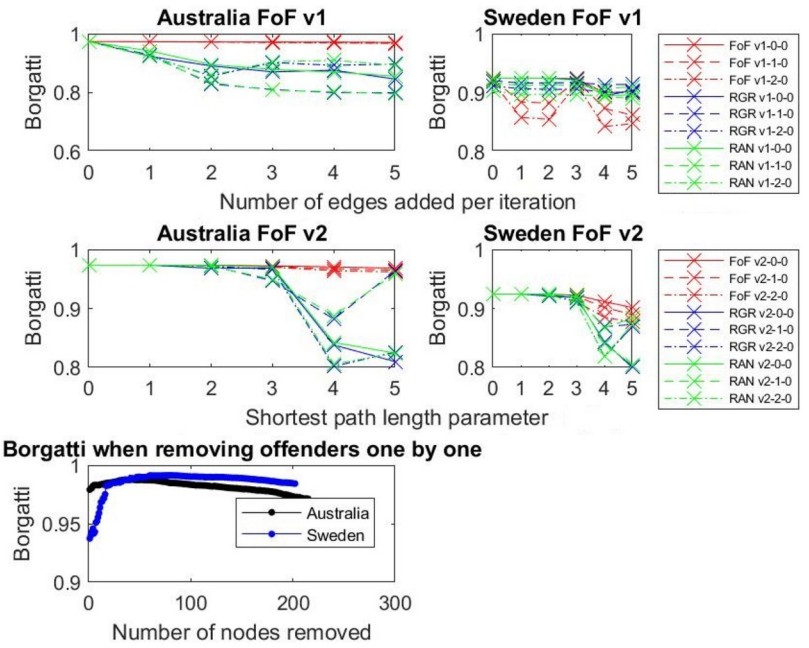

**Fig 10. Borgatti for different experiment settings.** The German data is very sparse, so the results are not presented here.

smaller clustering coefficient does not mean that this network is not small-world anymore. We ensure the small-world phenomenon by adding additional connections according to the friend-of-a-friend phenomenon, as proposed in FoF v1 and FoF v2. Given how the clustering coefficient is defined (number of triples connected to a given node divided by the number of all possible triples connected to this node), adding edges bridging women and men will initially cause the clustering coefficient to decrease (Fig 8) but more parts of the networks will be connected. This will ultimately result in building bridges between isolated groups and, as a result, enhance the small-world phenomenon.

**Borgatti** for FoF and removal of nodes (REM) does not change the network much, suggesting that this approach is not the best strategy for opening the network.

**Degree Assortativity goes down for Australian network** (it was the highest for Australian network) as we add more relationships–this indicates that as nodes with different degrees start mixing together, heterogeneity of the network is increased. This is not evident for the German and Swedish networks due to the fact that those networks are very sparse, and their clustering coefficient is very low–thus, they are random in their nature, while the Australian network is closer to a small-world model. Therefore, adjusting assortativity for the German and Swedish networks is not meaningful as the networks are too sparse.

The networks of all three industries (Australian, German, and Swedish) are very fragmented–there are many cliques that are isolated (Table 2). Removing the highly connected male producers (REM) means further fragmenting a largely fragmented network. FoF works locally, so it helps to diversify the network (there is a drop in degree assortativity in the case of the Australian network, for example) and break the tightly connected cliques (a drop in modularity and clustering coefficient). RAN and RGR work globally, so they can bridge the fragmented parts of the network, and this results in the further opening of the network (this can be seen in the drop of all metrics including the Borgatti fragmentation coefficient). Thus, the next natural step for analysing and opening these networks would be to create more advanced mechanisms that would bring the local- and global-level control mechanisms together.

## Shortcomings of the approach

In an attempt to represent network openness in terms of network science, we propose to combine two approaches: (i) one for addressing inequality as a type of network marginalisation (Lutter's network openness) and (ii) a second for building a good network of collaborations (small-world phenomenon). To achieve Lutter's recommendation for network openness we suggest decreasing modularity and assortativity. For building a healthy collaborative network we suggest promoting small-world behaviour where friend-of-my-friend is my friend. The limitation resulting from our approach is a lack of proof of the external validity of these measures in the same way as demonstrated by Lutter and this will be one of the future directions for this study.

When looking at our proposed control mechanisms, they are based on either removing nodes or adding new connections. However, the dynamics and evolution of social networks are much more complex than these mechanisms. Thus, future work will apply control mechanisms that are more sophisticated and able to simultaneously add/remove both nodes and relationships. Having said this, research concepts and results presented in this study are key to understanding how individual control mechanisms influence network dynamics before any ensemble approaches are investigated.

## Conclusion and future work

Our findings confirm those of Burt [45] and Ibarra [46], which demonstrate that, unless women can create ties to key players, they will remain on the periphery of the network.

Proposed control mechanisms give an indication of how this can be achieved. At the same time, we are mindful that, although our proposed interventions into male-dominated networks are conceptually transparent, they lack external validity as real-world strategies. For example, establishing employment connections between key male players in the film industry and women may not always be desirable for those women and could place them in difficult or damaging workplace situations.

Although more longitudinal research on the impact of change mechanisms in the film industry, such as gender equity policies, is warranted, it appears that the strength and importance of closed social networks in these sectors do act as a barrier to more diverse employment outcomes. In this context, it is possible to argue that the limited type and shortage of data used to research sectors such as the film industry both reflects and contributes to the persistence of ongoing inequities. Aggregated statistics describing the disproportional participation rates of women are useful only as snapshots of industrial injustices that are already well known. Making a numerical case for redressing gender inequality in the film industries will not create meaningful change. Gendered *social* relations are the root cause of inequality and we need to attune our research to understanding these dynamics.

As the natural next step for this research, we plan to build ensemble control mechanisms, taking into account both local (FoF) and global-level (RGR) approaches. We also need to consider natural dynamics together with the designed control mechanisms. We plan to start with random dynamics as the natural, as this is the easiest one to trace analytically. Another possibility is to enhance the simulations by targeting not only men who do not work with women but also their "friends of friends" (FoF). We also intend to work with cultural policy experts to understand the applied aspects of this research approach.

## Acknowledgments

The authors wish to acknowledge the collegiate support received from the extended membership of the Kinomatics Project in the development and execution of this research.

## Author Contributions

**Conceptualization:** Deb Verhoeven, Katarzyna Musial, Stuart Palmer.

**Data curation:** Deb Verhoeven, Stuart Palmer, Sarah Taylor, Shaukat Abidi, Vejune Zemaityte, Lachlan Simpson.

**Formal analysis:** Deb Verhoeven, Katarzyna Musial, Stuart Palmer, Sarah Taylor, Shaukat Abidi, Lachlan Simpson.

**Funding acquisition:** Deb Verhoeven, Katarzyna Musial.

**Investigation:** Deb Verhoeven, Katarzyna Musial, Stuart Palmer, Sarah Taylor.

**Methodology:** Deb Verhoeven, Katarzyna Musial, Stuart Palmer, Sarah Taylor.

**Project administration:** Deb Verhoeven.

**Resources:** Deb Verhoeven.

**Supervision:** Deb Verhoeven, Katarzyna Musial.

**Validation:** Deb Verhoeven, Katarzyna Musial, Stuart Palmer, Sarah Taylor.

**Visualization:** Deb Verhoeven, Katarzyna Musial, Stuart Palmer, Sarah Taylor.

**Writing – original draft:** Deb Verhoeven, Katarzyna Musial, Stuart Palmer.

**Writing – review & editing:** Deb Verhoeven, Katarzyna Musial, Stuart Palmer, Sarah Taylor, Shaukat Abidi, Vejune Zemaityte, Lachlan Simpson.

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
