## [Decision Letter · Decision Letter 0]

12 Feb 2020

PONE-D-19-29292

Controlling for optimal openness in the male-dominated collaborative networks of the global film industry

PLOS ONE

Dear Dr Verhoeven,

Thank you for submitting your manuscript to PLOS ONE. After careful consideration, we feel that it has merit but does not fully meet PLOS ONE’s publication criteria as it currently stands. Therefore, we invite you to submit a revised version of the manuscript that addresses the points raised during the review process.

* Revise the title which currently is misleading

* Clarify the new contributions from the current manuscript, particularly in the abstract

* Provide more information about the collection of the data and try to make some version of the data available

We would appreciate receiving your revised manuscript by Mar 28 2020 11:59PM. To enhance the reproducibility of your results, we recommend that if applicable you deposit your laboratory protocols in protocols.io, where a protocol can be assigned its own identifier (DOI) such that it can be cited independently in the future. For instructions see: http://journals.plos.org/plosone/s/submission-guidelines#loc-laboratory-protocols

We look forward to receiving your revised manuscript.

Kind regards,

Luís A. Nunes Amaral, Ph.D.

Academic Editor

PLOS ONE

Journal Requirements:

2. The manuscript indicates data curation and collection procedures can be found at the kinomatics website, but the linked reference leads to the general webpage and not the specifics of this study. In the manuscript, please include additional information about which data were collected for the study and what methods of collection were employed. If any websites were accessed to acquire this data ensure you have included a statement specifying whether the collection method complied with the terms and conditions for the website.

Reviewers' comments:

Reviewer's Responses to Questions

**Comments to the Author**

1. Is the manuscript technically sound, and do the data support the conclusions?

Reviewer #1: Partly

2. Has the statistical analysis been performed appropriately and rigorously? 

Reviewer #1: Yes

3. Have the authors made all data underlying the findings in their manuscript fully available?

Reviewer #1: No

4. Is the manuscript presented in an intelligible fashion and written in standard English?

Reviewer #1: Yes

5. Review Comments to the Author

Reviewer #1: In this manuscript, the authors study gender inequalities in the film industry. In order to understand its causes and come up with possible interventions, the authors argue to consider the network of relationships between individuals (producers, etc.). Looking at data from 3 datasets and using numerical simulations, the authors identify possible mechanisms to increase openness in networks.

I am convinced by the authors' argument that looking at the interaction networks of the producers will yield a better understanding of the possible causes of gender inequality. Therefore, I believe the current manuscript constitutes a genuine contribution and it is well written. However, before recommending its publication I would like the authors to address/clarify the following points:

1. In the abstract, the authors state that "we suggest that a study of network relationships offers innovative recommendations for understanding and redressing inequality in these industries.". I find a chance that this statement might be misleading with respect to what is done in the paper. It could be interpreted that the current paper is the first one to attempt a network approach; however, later (line 177) the authors write that "Lutter, who argues that the critical index of equity outcomes is network openness" showing that others have suggested similar things before. Furthermore, the following statement "Inequality is exacerbated when industry networks are most closed" suggests that it is a finding of the current paper; as I understand it is also a known result from Lutter. Clarifying the new contributions from the current manuscript, particularly in the abstract, would sharpen the manuscript and makes it easier for the reader to judge the importance of the paper.

2. I am slightly skeptical towards the formalization of Lutter's network openness in terms of small world (clustering coefficient, average shortest path), modularity, and assortativity. While these measures are common in studies of networks (and many networks are small-world-like and modular and assortative) we lack a proof of the external validity of these measures in the same way as demonstrated by Lutter. I am aware that these would probably require additional experiments which are beyond the scope of the current study. Thus, mentioning such limitations would make it easier to identify the open problems.

3. How is the openness of the networks measured? In the abstract, it is stated that "the most critical factor for improving network openness is ...". From the paper, I understood that (line 210) "we define an open network as one that follows a small-world model and at the same time aims at reducing modularity and node-degree assortativity." The interventions decrease the latter two. However, they also decrease the clustering coefficient, which makes it less small-world and thus less open. In view of the lack of a (more) clear definition of network openness, I find the conclusion not quantitatively supported by the analysis. The title ("optimal openness") is misleading as the relation between the intervention and openness is qualitative.

4. I was hoping to find more information about the collection of the data (page 10), for example what sources/films/genres were included or excluded and for what reasons. This would be interesting to assess whether the differences in the network characteristics (Table 2) could be due to the choices made in the data collection/pre-processing. Regarding the sharing of the data: I understand that the underlying data is sensitive and that it cannot be shared. However, in order to reproduce the analysis (and build upon it), would it be possible to publish partial or anonymized datasets, for example simply the network and the gender of the node (without any further meta information)?

6. PLOS authors have the option to publish the peer review history of their article (what does this mean?). If published, this will include your full peer review and any attached files.

Reviewer #1: No

---

## [Author Response · Author response to Decision Letter 0]

6 May 2020

We have addressed the reviewer comments in detail in our Letter of Rebuttal. Our response to the short summary of suggestions in the email is here: 

* Revise the title which currently is misleading

We have removed the word “optimal” in the title

* Clarify the new contributions from the current manuscript, particularly in the abstract

To emphasize the contributions of our work, we included in the abstract the following statement: “applying network analysis to understand inequality in the film industry has been undertaken. However, in this study we offer a comprehensive approach that enables us to not only understand what inequality in film industry looks like through the lens of network science but also how we can attempt to address this issue. We offer a data-driven simulation framework that investigates various what-if scenarios when it comes to network evolution. We then assess each of these scenarios with respect to its potential to address gender inequality in film industry.

* Provide more information about the collection of the data and try to make some version of the data available

More detailed information is now provided via correctly detailed references (the specific webpage is now identified as: https://kinomatics.com/the-gender-offender-analysis-how-and-why-we-did-it-part-one/

The network data is now publicly available on figshare with a DOI: https//doi.org/10.6084/m9.figshare.11959221.v1

*We have also removed the "&" symbol from the list of authors

---

## [Decision Letter · Decision Letter 1]

28 May 2020

Controlling for openness in the male-dominated collaborative networks of the global film industry

PONE-D-19-29292R1

Dear Dr. Verhoeven,

We are pleased to inform you that your manuscript has been judged scientifically suitable for publication and will be formally accepted for publication once it complies with all outstanding technical requirements.

With kind regards,

Luís A. Nunes Amaral, Ph.D.

Academic Editor

PLOS ONE

Additional Editor Comments (optional):

Reviewers' comments:

Reviewer's Responses to Questions

**Comments to the Author**

1. If the authors have adequately addressed your comments raised in a previous round of review and you feel that this manuscript is now acceptable for publication, you may indicate that here to bypass the “Comments to the Author” section, enter your conflict of interest statement in the “Confidential to Editor” section, and submit your "Accept" recommendation.

Reviewer #1: All comments have been addressed

2. Is the manuscript technically sound, and do the data support the conclusions?

Reviewer #1: Yes

3. Has the statistical analysis been performed appropriately and rigorously? 

Reviewer #1: Yes

4. Have the authors made all data underlying the findings in their manuscript fully available?

Reviewer #1: Yes

5. Is the manuscript presented in an intelligible fashion and written in standard English?

Reviewer #1: Yes

6. Review Comments to the Author

Reviewer #1: I want to thank the authors for their detailed and thorough response. The changes in the revised manuscript address all the main points I had raised in my previous comments leading to a subtantial improvement of the paper. Specifically, the authors i) changed the abstract to clarify the contributions of the paper, ii) extended a discussion around the limitations of the external validity of the network measures, iii) clarified the trade-off in openness between high clustering-coefficent (small world) and small modularity; in particular by changing the title, and iv) added information about the data collection and shared the data in figshare. Therefore, I now recommend publication of the papaer in Plos One.

7. PLOS authors have the option to publish the peer review history of their article (what does this mean?). If published, this will include your full peer review and any attached files.

Reviewer #1: Yes: Martin Gerlach

---

## [Editor Report · Acceptance letter]

3 Jun 2020

PONE-D-19-29292R1 

Controlling for openness in the male-dominated collaborative networks of the global film industry 

Dear Dr. Verhoeven:

I'm pleased to inform you that your manuscript has been deemed suitable for publication in PLOS ONE. Congratulations! Your manuscript is now with our production department. 

Kind regards, 

on behalf of

Dr. Luís A. Nunes Amaral 

Academic Editor

PLOS ONE